# LEARNING TO REASON OVER CONTINUOUS TOKENS WITH REINFORCEMENT LEARNING

**Yiran Zhao    Yuhui Xu    Doyen Sahoo    Caiming Xiong    Junnan Li**

Salesforce AI Research

## ABSTRACT

Large Language Models (LLMs) have shown strong performance in complex reasoning tasks, especially when guided by Chain-of-Thought (CoT) prompting. However, conventional CoT reasoning in the discrete token space suffers from high computational and memory costs due to verbose intermediate steps. Recent work has explored latent reasoning in the embedding space to improve efficiency, but often at the cost of clarity and performance. In this work, we propose Hybrid Reasoning (`HyRea`), a unified framework that enables LLMs to dynamically switch between explicit (token-based) and latent (embedding-based) reasoning during inference. To train the model to make these decisions effectively, we introduce a two-stage training pipeline: (1) a supervised cold-start phase that introduces latent reasoning by replacing low-entropy CoT steps with embeddings, and (2) a reinforcement learning phase using Group Relative Policy Optimization (GRPO) to fine-tune the model's reasoning strategy based on task-specific rewards. Experiments on mathematical reasoning benchmarks show that `HyRea` achieves significant reductions in token usage while maintaining or improving accuracy, offering an effective and scalable solution for efficient multi-step reasoning in LLMs. Our code is publicly available at `https://github.com/zhaoyiran924/HyRea`.

## 1 INTRODUCTION

Large Language Models (LLMs) (OpenAI, 2023; Grattafiori et al., 2024; Team et al., 2023; 2024; Yang et al., 2024a; Comanici et al., 2025; OpenAI, 2025) have shown impressive capabilities in complex reasoning tasks (Wang et al., 2024; Hsiao et al., 2025; Shi et al., 2025; Qu et al., 2025), particularly when guided by Chain-of-Thought (CoT) prompting (Wei et al., 2022), which encourages step-by-step intermediate reasoning. However, conventional CoT reasoning operates entirely in the discrete token space, leading to inefficiencies in computation and memory due to verbose outputs. As long-context and cost-effective inference become increasingly important, especially in reinforcement learning (RL) settings for math tasks (Liu et al., 2024), methods like Deepseek-R1 (Guo et al., 2025) face high token costs and slow convergence. To tackle these challenges, recent approaches propose operating directly in the embedding space, bypassing tokenization altogether (Hao et al., 2024b; Yue et al., 2025; Zhang et al., 2025). These latent reasoning methods offer substantial token compression and improved efficiency. However, reasoning over embeddings remains inherently difficult and can lead to degraded model performance compared to traditional token-based inference, as certain tokens encode complex, nuanced information that cannot be fully preserved in compressed embeddings (Hao et al., 2024b; Shen et al., 2025). Moreover, current models lack the capability to selectively determine which tokens should be compacted and which should remain in their original form.

In this work, we propose a novel Hybrid Reasoning (`HyRea`) framework that combines explicit (token-based) and latent (embedding-based) reasoning[1]. Explicit reasoning offers interpretability through step-by-step generation but is inefficient, while latent reasoning improves efficiency by operating in embedding space, often sacrificing clarity and performance. As shown in Figure 1, `HyRea` enables LLMs to dynamically switch between these two modes during inference, selecting the most suitable reasoning strategy based on the context. This adaptive mechanism allows the model to maintain high accuracy while significantly reducing the number of generated tokens, yielding

---

[1]Throughout this paper, we use the term "embedding" to refer specifically to the last-layer hidden state.

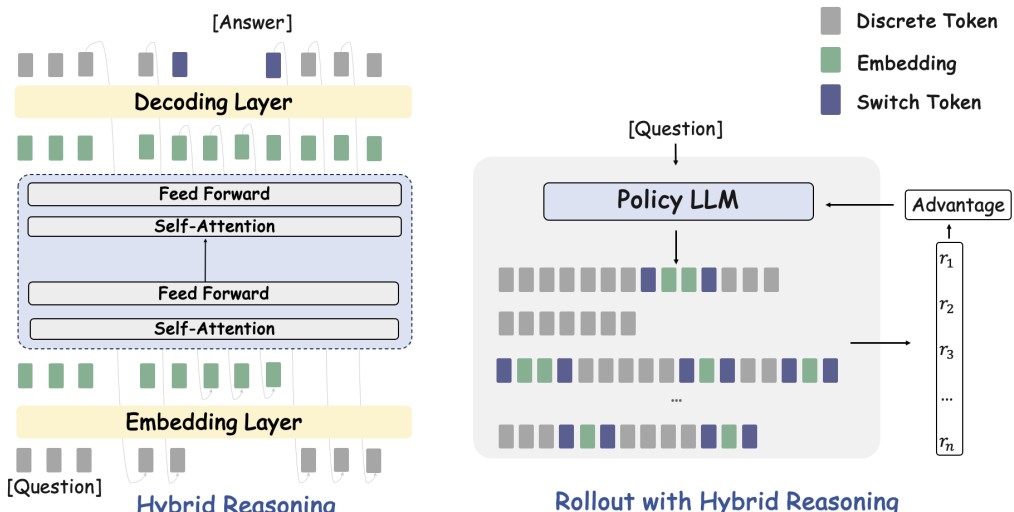

Figure 1: `HyRea` framework overview. The model dynamically switches between explicit (token-based) and latent (embedding-based) reasoning during inference. Explicit reasoning provides interpretability through step-by-step token generation, while latent reasoning operates in embedding space for greater efficiency. `HyRea` adaptively selects the optimal reasoning mode based on context, enabling a flexible and efficient hybrid approach.

better trade-offs between performance and efficiency. To support this hybrid reasoning capability, we introduce a two-stage training framework. In the *supervised cold-start phase*, the model learns to use latent reasoning by partially replacing intermediate CoT steps with continuous embeddings. To guide this replacement, we prioritize steps with low entropy, which are more deterministic and thus easier for the model to learn in latent space. This encourages the model to interpret and generate latent computations within a broader reasoning trajectory. In the subsequent *reinforcement learning phase*, to further refine the model's decision-making process. Specifically, we adopt Group Relative Policy Optimization (GRPO) (Shao et al., 2024), a RL technique that stabilizes learning by evaluating policies within sampled groups. This phase enables the model to learn when to invoke explicit versus latent reasoning based on downstream rewards, such as accuracy, format correctness, and efficiency. By integrating explicit and latent reasoning in a unified and learnable framework, `HyRea` provides a flexible and scalable approach to improve reasoning performance in LLMs, particularly in domains like mathematics where both precision and efficiency are critical.

We conduct comprehensive experiments to evaluate the performance of `HyRea` across various models and tasks. Our compression method was evaluated on two widely used open-source models, Qwen2.5-7B-Instruct and Qwen2.5-32B-Instruct (Yang et al., 2024a), and we demonstrate that it can reduce length of the answer to approximately 60% of its original length without any loss in mathematical reasoning capability. Specifically, on MATH-500 (Lightman et al., 2023), Minerva Math (Hendrycks et al., 2021), AMC23 (AMC, 2023), and Olympiad Bench (He et al., 2024), `HyRea` achieves competitive accuracies while generating solutions with an average of only 309, 419, 452, and 492 tokens under Qwen2.5-7B, respectively. In contrast, Qwen2.5-7B-Instruct trained using a conventional reinforcement learning approach produces substantially longer outputs, averaging 698, 671, 892, and 854 tokens on the same benchmarks. Furthermore, we evaluate the model trained with `HyRea` on general tasks beyond mathematics, including MMLU (Hendrycks et al., 2020) and GPQA (Rein et al., 2024). The results demonstrate strong generalization ability and a significant reduction in output length.

## 2 INTEGRATE EXPLICIT AND LATENT REASONING

### 2.1 EXPLICIT REASONING

Explicit reasoning refers to the process by which a model generates reasoning outputs in an autoregressive manner, decoding one token at a time. Formally, given an input query $x = [x_1, x_2, \cdots, x_t]$,

the model first maps each token to its corresponding embedding via the embedding layer, resulting in the sequence $E = [e(x_1), e(x_2), \cdots, e(x_t)]$. These embeddings are then processed by a series of Transformer blocks as introduced by Vaswani et al. (2017):

$$H_t = [h_1, h_2, \cdots, h_t] = \text{Transformer}(E), \tag{1}$$

where $h_i$ denotes the hidden embeddings at the $i$-th position from the final layer of the Transformer. Furthermore, the hidden state $h_t$ is transformed into a probability distribution over the vocabulary set $\mathcal{V}$ through the language modeling head, denoted as $\text{LM}_{\text{head}}$. The next token $\hat{x}_{t+1}$ is then selected by taking the argmax over the resulting logits:

$$\hat{x}_{t+1} = \arg\max_{\mathcal{V}}(\text{logits}) = \arg\max_{\mathcal{V}} \left( \text{LM}_{\text{head}}(h_t) \right). \tag{2}$$

The input is then updated to $[x_1, x_2, \cdots, x_t, \hat{x}_{t+1}]$, which is subsequently processed sequentially by Equation 1 followed by Equation 2.

## 2.2 Latent Reasoning

Instead of decoding token by token, Chain of Continuous Thought (Coconut) (Hao et al., 2024b) proposes to bypass tokenization entirely by operating directly on embeddings. Specifically, during decoding, the model feeds the hidden output from the final layer back into the first layer as input. That is, rather than decoding the next token using Equation 2, the model proceeds as follows:

$$H_{t+1} = [h_1, h_2, \cdots, h_t, h_{t+1}] = \text{Transformer}(E \parallel h_t), \tag{3}$$

where $E$ denotes the initial embedding sequence and $h_t$ is the hidden state at time step $t$ and $\parallel$ represents concatenation. Compared to explicit reasoning, operating directly on embeddings allows for a more compact representation during inference, potentially reducing token consumption. However, reasoning over embeddings remains inherently difficult and can lead to degraded model performance compared to traditional token-based inference, as certain tokens encode complex, nuanced information that cannot be fully preserved in compressed embeddings. This loss of semantic fidelity can be particularly detrimental in tasks that require precise symbolic manipulation, such as mathematical reasoning or code generation, where even small distortions in representation may lead to incorrect conclusions. Moreover, current models lack the capability to selectively determine which tokens should be compacted and which should remain in their original form. Compression is typically applied uniformly or based on fixed heuristics, without accounting for the varying informational value of different tokens within a reasoning trace. This inability to adaptively balance compression and fidelity limits the effectiveness of latent reasoning and highlights the need for more principled, content-aware strategies for selective representation.

## 2.3 Hybrid Reasoning

To improve the efficiency of the reasoning process while minimizing performance degradation, we aim for the model to flexibly switch between explicit and latent reasoning during inference. When decoding the next position, the model can autonomously decide whether to operate in the embedding space or the token space—that is, whether to perform latent reasoning or explicit reasoning. Formally, at position $i$, if the model chooses to employ the explicit reasoning mode, it follows Equation 2 to generate the next token. Conversely, if the model opts for the latent reasoning mode, it applies Equation 3, inserting a `<start-latent>` token at the beginning and a `<end-latent>` token at the end to mark the latent reasoning span. Formally, the ideal structure of hybrid reasoning can be represented as:

$$[\text{Question}] \, [\text{Step}_1] \cdots \texttt{<start-latent>} \, [\texttt{latent}] \, \texttt{<end-latent>} \cdots [\text{Step}_N] \cdots [\text{Answer}].$$

## 3 Training Method

In this section, we mainly introduce the training method to enable the model conduct hybrid reasoning.

---

**Algorithm 1** Hybrid Reasoning Training (`HyRea`)

---

**Input:** Pretrained language model $\mathcal{LLM}$, CoT training data $\mathcal{D}_{\text{SFT}}$, RL training data $\mathcal{D}_{\text{RL}}$, max latent steps $S$, group size $G$, clipping threshold $\varepsilon$
**Output:** Hybrid reasoning model $\mathcal{LLM}$
 1: // Stage 1:  Cold-Start Supervised Fine-Tuning
 2: **for** $s$ from 1 to $S$ **do**
 3:     **for** each batch $(q, a)$ in $\mathcal{D}_{\text{SFT}}$ **do**
 4:         Identify reasoning steps with low entropy
 5:         Replace $s$ selected steps with $[$`<start-latent>` latent `<end-latent>`$]$
 6:         Compute loss $\mathcal{L}_{\text{SFT}} = -\log \mathcal{LLM}(q \setminus [\text{latent}])$
 7:         Update $\mathcal{LLM}$ using gradient descent on $\mathcal{L}_{\text{SFT}}$
 8:     **end for**
 9:     Incrementally introduce 10% new data into $\mathcal{D}_{\text{SFT}}$ at each iteration
10: **end for**
11: // Stage 2:  Reinforcement Learning with GRPO
12: **for** each query $q$ in $\mathcal{D}_{\text{RL}}$ **do**
13:     Sample $G$ outputs $\{o_1, o_2, \ldots, o_G\} \sim \mathcal{LLM}_{\text{old}}(\cdot|q)$
14:     Evaluate rewards $\{r_1, \ldots, r_G\}$ (accuracy, formatting, latent usage)
15:     Compute GRPO loss by Equation 6
16:     Update $\mathcal{LLM}$ using gradient descent on $\mathcal{L}_{\text{GRPO}}$
17: **end for**

---

### 3.1 COLD START FOR REPLACE LOW-ENTROPY STEPS

To enable the model to perform hybrid reasoning, we adopt a cold-start approach inspired by Deepseek-R1 (Liu et al., 2024). In this initial phase, the model is trained using supervised fine-tuning (SFT) on chain-of-thought (CoT) (Wei et al., 2022) reasoning data to learn preliminary switching capabilities. This serves as a foundational step toward developing more advanced hybrid reasoning abilities. Specifically, the original CoT data is structured as a sequence:

$$C := [\text{Question}], [\text{Step}_1], [\text{Step}_2], \ldots, [\text{Step}_N], [\text{Answer}]. \tag{4}$$

To introduce latent reasoning, we select steps that have low entropy and replace them with a latent segment $[\text{Latent}] := $ `<start-latent>` $c \times [\text{latent}]$ `<end-latent>`, where $c$ denotes the number of latent tokens. This creates a hybrid sequence in which $[\text{Latent}]$ serves as a placeholder for latent reasoning. Importantly, the entropy-based replacement strategy prevents the model from compacting tokens that encode complex or critical information that cannot be faithfully represented by latent embeddings. During training, the loss is computed only over the visible (non-latent) tokens, ensuring that the model focuses on learning the interpretable parts of the reasoning trace while implicitly handling the latent segments.

$$\mathcal{L}_{\text{cold start}} := -\log \mathcal{LLM}(C \setminus [\text{Latent}]), \tag{5}$$

where $\mathcal{LLM}(\cdot)$ denotes the language model's likelihood function. In addition, the number of steps replaced by $[\text{Latent}]$ increases progressively during training, starting from 0 and gradually reaching a predefined maximum threshold $S$.

### 3.2 REINFORCEMENT LEARNING

We mainly employ the Group Relative Policy Optimization (GRPO) proposed by Shao et al. (2024)

$$\mathcal{L}_{\text{GRPO}}(\theta) = \mathbb{E}_{q \sim P(Q), \{o_i\}_{i=1}^G \sim \pi_{\theta_{\text{old}}}(O|q)}$$
$$\left[ \frac{1}{G} \sum_{i=1}^G \left( \min \left( \frac{\pi_\theta(o_i \mid q)}{\pi_{\theta_{\text{old}}}(o_i \mid q)} A_i, \text{clip} \left( \frac{\pi_\theta(o_i \mid q)}{\pi_{\theta_{\text{old}}}(o_i \mid q)}, 1 - \varepsilon, 1 + \varepsilon \right) A_i \right) \right) \right], \tag{6}$$

where

$$A_i = \frac{r_i - \text{mean}(\{r_1, r_2, \cdots, r_G\})}{\text{std}(\{r_1, r_2, \cdots, r_G\})}, \tag{7}$$

and reward consists of both a format reward and an accuracy reward, as well as a latent reward. Specifically, we use the latent reward to guide the model in generating the token [Latent]. Additionally, during loss computation, we exclude the [Latent] token and calculate the loss only over the remaining token steps. The overall algorithm is further illustrated in Algorithm 1. Through reinforcement learning, this approach eliminates the need for manually constructing synthetic data to train the model's switching capability. Instead, the model learns to perform hybrid reasoning in a self-supervised manner, guided by reward signals that reflect both correctness and efficiency.

## 4 EXPERIMENT

### 4.1 EVALUATION SETUP

**Datasets** We primarily utilize two large-scale mathematical reasoning datasets for cold start: NuminaMath-CoT (LI et al., 2024), which contains 860k diverse math problems ranging from high school exercises to international olympiad-level questions formatted in a CoT style, and MetaMathQA (Yu et al., 2023), comprising 390k problem-solution pairs enhanced through various data augmentation techniques to promote diverse and robust reasoning pathways. Specifically, we split the answer by '$\backslash n$' and '.', and ensure that each equation is kept as an independent step. Figure 2 presents a concrete example of the processed dataset. Additionally, in the reinforcement learning stage, we adopt Math-12k[2], a challenging math dataset derived from (Lightman et al., 2023).

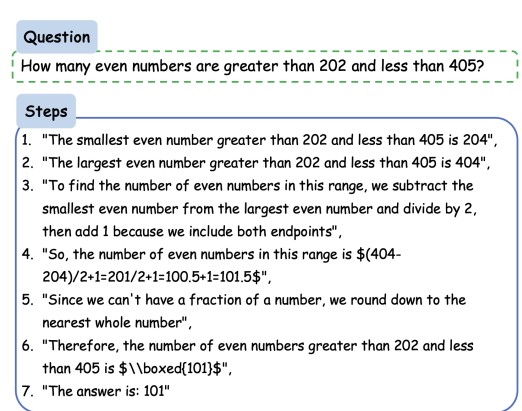

Figure 2: An example data point.

**Backbone Models** We evaluate two widely-used open-source LLMs as backbone models. Qwen2.5-7B-Instruct and Qwen2.5-32B-Instruct (Yang et al., 2024a).

**Baselines.** We employ several state-of-the-art efficient reasoning method with less tokens (i) CoT (Wei et al., 2022); (ii) Reinforcement Learning (Guo et al., 2025), which optimizes reasoning strategies through trial-and-error by rewarding accurate outputs; (iii) Chain of Continuous Thought (Coconut) (Hao et al., 2024b), a novel paradigm that replaces discrete tokens with latent "continuous thoughts" by feeding the last hidden state directly back into the model, allowing reasoning in an unconstrained latent space and enabling breadth-first exploration over multiple reasoning paths; (iv) Soft thinking (Zhang et al., 2025), a training-free method that mimics human-like soft reasoning by generating probability-weighted concept tokens in a continuous concept space, capturing semantic ambiguity and enabling more abstract, flexible reasoning with significantly fewer tokens.

**Benchmarks** To assess mathematical reasoning proficiency, we employ four benchmarks: MATH-500 (Lightman et al., 2023), a 500-question subset of the MATH benchmark curated for rigorous; Minerva Math (Hendrycks et al., 2021); Olympiad Bench (He et al., 2024) as well as competition-level benchmarks such as AMC23 (AMC, 2023). We evaluate the performance of the model on each dataset using `pass@1`. Additionally, we analyze the length of each generated answer by counting the total number of tokens, including both standard tokens and the [Latent] token. Furthermore, we assess the frequency of switches between latent and explicit reasoning within each response.

**Implement Details** For the cold-start, we employed the LLaMA-Factory library (Zheng et al., 2024), a widely adopted GitHub-hosted framework for efficient large-model fine-tuning, to carry out all training procedures. Experiments are conduced on eight 140GB NVIDIA H200 GPUs, with learning rate as $4 \times 10^{-7}$, global batch size as 32. Furthermore, to reduce memory consumption during training, we applied ZeRO Stage-2 optimization and gradient checkpointing, both provided

---

[2]https://huggingface.co/datasets/hiyouga/math12k

Table 1: Main results of `HyRea` across four math reasoning benchmarks. We report accuracy (Acc), average number of output tokens (#Tokens), and average number of latent-explicit switches (#Switch).

| | Methods | Math-500 | | | Minerva Math | | | AMC23 | | | Olympiad Bench | | |
|---|---|---|---|---|---|---|---|---|---|---|---|---|---|
| | | Acc | #Tokens | #Switch | Acc | #Tokens | #Switch | Acc | #Tokens | #Switch | Acc | #Tokens | #Switch |
| **Qwen2.5-7B-It** | SFT | 73.6 | 546 | 0 | 26.1 | 619 | 0 | 36.1 | 845 | 0 | 29.3 | 723 | 0 |
| | SFT + RL | 84.2 | 698 | 0 | 26.8 | 671 | 0 | 48.2 | 892 | 0 | 40.0 | 854 | 0 |
| | Coconut | 70.4 | 106 | 1 | 22.1 | 174 | 1 | 33.7 | 217 | 1 | 26.8 | 296 | 1 |
| | Soft Thinking | 66.4 | 617 | 1 | 16.9 | 604 | 1 | 24.1 | 784 | 1 | 24.7 | 595 | 1 |
| | `HyRea` w/o RL | 71.8 | 248 | 4.8 | 20.6 | 288 | 3.1 | 34.9 | 339 | 4.9 | 28.5 | 378 | 4.3 |
| | `HyRea` | 83.6 | 387 | 4.4 | 27.2 | 425 | 3.6 | 48.2 | 526 | 4.7 | 39.6 | 583 | 3.8 |
| **Qwen2.5-32B-It** | SFT | 83.6 | 595 | 0 | 37.1 | 644 | 0 | 55.4 | 803 | 0 | 49.0 | 866 | 0 |
| | SFT + RL | 85.2 | 588 | 0 | 39.7 | 608 | 0 | 61.4 | 905 | 0 | 49.5 | 899 | 0 |
| | Coconut | 79.8 | 179 | 1 | 32.4 | 209 | 1 | 53.0 | 285 | 1 | 46.2 | 403 | 1 |
| | Soft Thinking | 82.4 | 574 | 1 | 33.4 | 602 | 1 | 55.4 | 776 | 1 | 43.7 | 887 | 1 |
| | `HyRea` w/o RL | 79.8 | 305 | 3.2 | 34.6 | 267 | 3.9 | 54.2 | 304 | 5.2 | 47.3 | 428 | 4.7 |
| | `HyRea` | 84.4 | 369 | 4.1 | 38.6 | 381 | 3.7 | 57.8 | 498 | 4.9 | 48.9 | 563 | 5.3 |

by the DeepSpeed library. Note that in the cold-start stage, the loss on `<start-latent>` and `<end-latent>` are scaled by a factor of 4 to emphasize their importance and help the model learn when to switch. For reinforcement learning training, we use the EasyR1 (Zheng et al., 2025) framework built on verl (Sheng et al., 2024), with specialized support for VLMs. Experiments are conducted using eight 140GB NVIDIA H200 GPUs with a global batch size of 128, a rollout batch size of 128, a rollout temperature of 1.0, a consistent learning rate of $1 \times 10^{-6}$, and 8 rollouts.

## 4.2 MAIN RESULT

**`HyRea` achieves strong accuracy–efficiency trade-offs.** As shown in Table 1, our proposed hybrid reasoning framework `HyRea` consistently delivers strong performance across all four math reasoning benchmarks, demonstrating both high accuracy and efficient token usage. Under the Qwen2.5-7B-Instruct, `HyRea` achieves an accuracy of 83.6 on MATH-500, nearly matching the SFT+RL baseline (84.2), while using less than half the number of output tokens (387 vs. 698). Similar trends are observed across other benchmarks: on Minerva Math, `HyRea` slightly outperforms SFT+RL (27.2 vs. 26.8) with a substantial reduction in tokens (425 vs. 671); and for AMC23 and Olympiad Bench, it matches or closely approaches the best-performing baselines in accuracy (48.2 and 39.6, respectively), while reducing token usage by nearly 50%. When scaled up to Qwen2.5-32B-Instruct, `HyRea` continues to show competitive results, achieving 84.4 accuracy on MATH-500 with only 369 tokens, compared to 85.2 accuracy and 588 tokens for SFT+RL. On Minerva Math and AMC23, `HyRea` reaches 38.6 and 57.8 accuracy, respectively, again using significantly fewer tokens than the SFT+RL counterpart. Notably, `HyRea` also exhibits meaningful latent-explicit switching behavior, averaging 3.6 to 5.3 mode transitions per sample depending on the dataset and model size, compared to zero switches for all baselines and only one for Coconut and Soft Thinking. This switching capacity reflects `HyRea`'s ability to dynamically leverage both explicit and latent reasoning strategies. Importantly, `HyRea` outperforms token-efficient approaches like Coconut, which, despite producing shorter outputs (e.g., 106–179 tokens on MATH-500), suffers from significant accuracy degradation across all tasks. The ablation study (`HyRea` w/o RL) further highlights the benefits of reinforcement learning, as performance notably drops when RL is removed. Overall, these results underscore `HyRea`'s effectiveness as a scalable solution for multi-step reasoning—achieving high accuracy with compact solutions via strategic reasoning mode transitions.

Complementing this, Figure 3 shows that during training, the accuracy and latent rewards steadily improve and stabilize, while the format reward remains consistently high. This indicates that `HyRea` learns to balance correctness, structure, and reasoning mode usage effectively. Overall, these results underscore `HyRea`'s effectiveness as a scalable solution for multi-step reasoning—achieving high accuracy with compact solutions via strategic reasoning mode transitions.

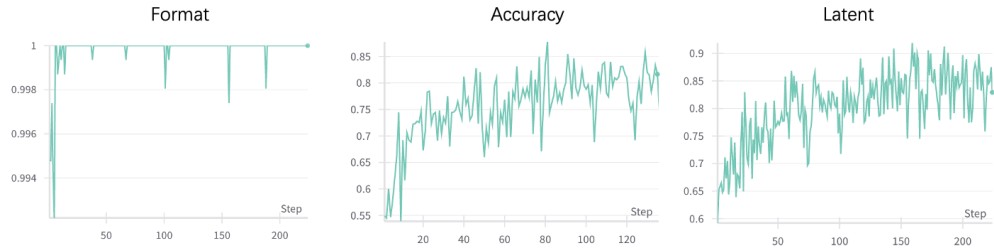

Figure 3: Reward over steps.

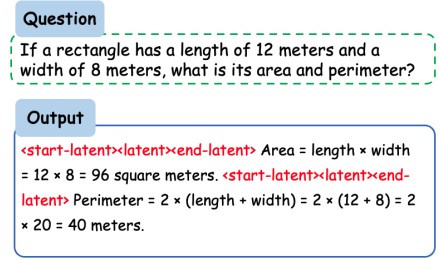

Figure 4: Concrete Example.

Table 2: Comparison of entropy based replacement and random sampling replacement.

| | Math | | Minerva | | AMC23 | | Olympiad | |
|---|---|---|---|---|---|---|---|---|
| | Acc | #Token | Acc | #Token | Acc | #Token | Acc | #Token |
| SFT+RL | 84.2 | 698 | 26.1 | 619 | 48.2 | 892 | 40.0 | 854 |
| Random | 83.4 | 309 | 26.5 | 419 | 49.4 | 452 | 39.6 | 492 |
| Entropy | 83.6 | 287 | 27.2 | 372 | 48.2 | 426 | 39.6 | 483 |

**Concrete Examples** Figure 4 exemplifies `HyRea`'s ability to abstract away low-utility, generic reasoning steps into latent space, effectively bypassing verbose yet semantically redundant content. By retaining only semantically salient computations and final outputs in explicit token space, the model achieves a more compact and efficient reasoning trace without compromising interpretability. This demonstrates HyRea's capacity to balance informativeness and conciseness through dynamic reasoning mode selection.

## 4.3 ABLATION ANALYSIS

**Random Replacement** Table 2 compares two strategies for selecting intermediate reasoning steps to replace with latent embeddings: random sampling versus entropy-based selection. Both strategies significantly reduce token usage compared to the CoT+RL baseline while achieving comparable or better accuracy. Notably, the entropy-based method consistently produces shorter outputs across all benchmarks, with token counts of 287 on MATH-500, 372 on Minerva Math, 426 on AMC23, and 483 on Olympiad Bench—representing the lowest token usage among all methods evaluated. In contrast, random replacement uses slightly more tokens (e.g., 309, 419, 452, and 492, respectively), suggesting that entropy-based selection produces more compressible reasoning paths. In terms of accuracy, entropy-based replacement also demonstrates greater stability. On Minerva Math, it achieves an accuracy of 27.2 compared to 26.5 for random, and on MATH-500 and Olympiad Bench, it matches or slightly outperforms random selection. Although both strategies come close to the CoT+RL performance in accuracy, entropy-based replacement offers a more favorable trade-off between performance and efficiency. These results validate our design choice of using entropy as a principled criterion for identifying deterministic and compressible reasoning steps to encode in latent space.

As shown in Figure 5, entropy-based sampling significantly outperforms random sampling during the cold-start stage. It achieves faster performance gains, surpassing 80 points within 10 iterations, while sample-based selection improves more slowly and plateaus around 75. This demonstrates that uncertainty-aware sampling more effectively prioritizes informative examples early in training, accelerating convergence and yielding better final performance.

**Replace Number of Latent** Figure 7 presents an ablation study on the number of latent replacements, showing how varying the parameter $c$ affects both accuracy and output length. As $c$ increases from 1 to 8, accuracy (purple line) drops sharply—from over 80% to below 10%—indicating that

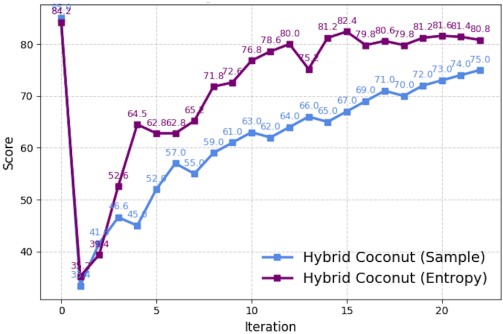

Figure 5: Cold-start stage: entropy-based vs. random sampling.

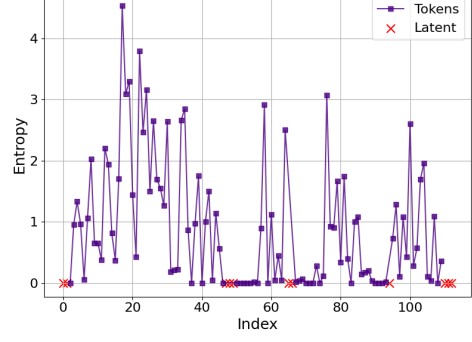

Figure 6: Concrete Example.

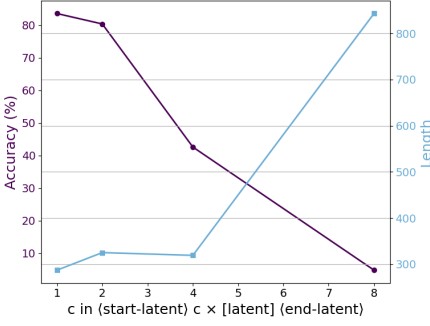

Figure 7: Ablation of replace number of latent.

Table 3: Generalization ability of `HyRea` trained model.

|  | **MMLU** | | **GPQA** | |
| --- | --- | --- | --- | --- |
|  | Acc | #Token | Acc | #Token |
| SFT | 70.2 | 84 | 23.7 | 837 |
| SFT+RL | 73.4 | 102 | 29.8 | 1083 |
| `HyRea` | 68.6 | 53 | 27.4 | 685 |

replacing more latent components substantially degrades model performance. Conversely, output length (blue line) increases significantly, particularly after $c = 4$, suggesting that excessive replacement leads to verbosity or redundancy. This inverse relationship between $c$ and performance suggests that the training stability of `HyRea` and hybrid reasoning deteriorates rapidly as $c$ increases.

## 5 FURTHER ANALYSIS

### 5.1 SWITCHING PATTERN

Figure 6 provides a fine-grained view of `HyRea`'s reasoning behavior via entropy over decoding steps. We observe that latent reasoning steps (marked with red crosses) tend to appear in low-entropy regions, indicating high model confidence in selecting the latent mode. Notably, these latent spans often occur at the beginning or end of the reasoning trajectory, suggesting `HyRea` prefers to compress either the problem setup or the final answer derivation when confident. Additionally, switching between modes is relatively infrequent, aligning with our earlier observation of around 3–5 switches per sample. However, each latent span typically contains multiple latent steps rather than isolated calls, indicating that `HyRea` strategically groups latent reasoning into longer, coherent segments. This behavior highlights `HyRea`'s ability to balance interpretability and efficiency by using explicit reasoning in uncertain mid-process steps, while leveraging latent modules for confident, compressed segments.

### 5.2 ASSESSING MODEL GENERALIZATION CAPABILITY

We evaluate the generalization ability of the `HyRea`-trained model on two benchmark datasets: MMLU (Hendrycks et al., 2020) and GPQA (Rein et al., 2024). As shown in Table 3, `HyRea` is compared against models trained with supervised fine-tuning (SFT) and reinforcement learning (SFT+RL). While SFT+RL achieves the highest accuracy (73.4 on MMLU and 29.8 on GPQA), it

generates substantially longer responses (102 and 1083 tokens, respectively). In contrast, `HyRea` produces more concise outputs (53 tokens on MMLU and 685 on GPQA) while maintaining competitive accuracy (68.6 and 27.4). This demonstrates `HyRea`'s strong generalization ability to untrained domains, balancing performance and efficiency without task-specific optimization. Such generalization is crucial for deploying language models in diverse, real-world settings where robustness and adaptability are essential.

## 6 RELATED WORKS

**Chain-of-Thought Reasoning** Chain-of-Thought (CoT) prompting (Wei et al., 2022) improves model performance by guiding the generation of intermediate reasoning steps. Such behavior can be elicited directly through prompt engineering without additional training (Khot et al., 2022; Zhou et al., 2022). To further enhance reasoning quality, recent works employ supervised fine-tuning and reinforcement learning to explicitly optimize multi-step reasoning (Yue et al., 2023; Yu et al., 2023; Wang et al., 2023; Xiong et al., 2025). An important extension of CoT involves integrating it with tree search algorithms (Yao et al., 2023; Hao et al., 2024a; Xie et al., 2023; Liao et al., 2025b), allowing models to explore and evaluate multiple reasoning paths to achieve better performance on complex tasks. These advances have been formalized into test-time inference scaling laws (Snell et al., 2024; Wu et al., 2024), which provide theoretical grounding for the observed performance improvements. The success of advanced reasoning models such as OpenAI's O1 (OpenAI et al., 2024) and DeepSeek's R1 (DeepSeek-AI et al., 2025) has further intensified interest in test-time search (TTS) techniques. However, the deliberative nature of these methods often leads to inefficiencies in token usage. As a result, reasoning efficiency has become an increasingly important research focus (Liu et al., 2025; Xu et al., 2025; Liao et al., 2025a; Aggarwal & Welleck, 2025).

**Latent Reasoning** Recent advances in LLM reasoning in latent space (Yang et al., 2024b; Biran et al., 2024) underscore the significance of hidden computations. To better leverage these latent dynamics, a range of methods introduce special tokens to explicitly guide reasoning. For instance, *<pause>* tokens Goyal et al. (2023) are incorporated during both pretraining and downstream finetuning, yielding consistent improvements over standard training. Similarly, non-semantic filler tokens (e.g., '...') Pfau et al. (2024) have been shown to enhance performance on certain reasoning tasks. Several works explore alternatives to explicit chain-of-thought (CoT) prompting. Yu et al. (2024) proposes implicit CoT, where models internalize intermediate reasoning steps distilled from explicit CoT supervision. Wang et al. (2023) introduces a hierarchical reasoning structure guided by planning tokens, which demarcate latent reasoning stages. More recently, Hao et al. (2024b) proposes COCONUT, replacing discrete CoT tokens with continuous latent embeddings, leading to improved reasoning across multiple tasks. Zhu et al. (2025) provides theoretical justification for the superiority of continuous CoTs over their discrete counterparts. In contrast to these approaches, our work focuses on selectively replacing low-entropy tokens with continuous embeddings, allowing the model to dynamically switch between discrete and continuous reasoning modes. We formulate this switching behavior as a reinforcement learning problem, enabling the model to learn when and how to invoke latent computation for improved reasoning quality.

## 7 CONCLUSION

We presented `HyRea`, a hybrid reasoning framework that allows large language models to dynamically alternate between explicit token-based reasoning and latent embedding-based reasoning, aiming to balance interpretability, accuracy, and efficiency. To train this capability, we proposed a two-stage approach that begins with entropy-guided supervised fine-tuning, where low-entropy reasoning steps are selectively replaced with latent representations, followed by reinforcement learning using Group Relative Policy Optimization to refine the model's reasoning strategy based on task-specific rewards. Extensive experiments on challenging mathematical benchmarks demonstrate that `HyRea` achieves substantial reductions in output length while maintaining or improving accuracy compared to strong baselines. Furthermore, `HyRea` exhibits strong generalization ability on non-mathematical tasks, showing its potential as a versatile and scalable solution for efficient multi-step reasoning in large language models.

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

## LLM USAGE

We used LLMs as general-purpose writing and debugging assistants. Specifically, LLMs were employed to help polish the writing (e.g., improving sentence clarity, grammar, and flow) and occasionally to assist with debugging minor implementation issues (e.g., identifying syntax errors or suggesting code refactoring). However, all core ideas, research questions, methodological designs, codebase implementations, experiments, and analyses were entirely conceived, developed, and conducted by the authors. No part of the intellectual contribution, experimental framework, or scientific reasoning was generated by an LLM.

## LIMITATION

While `HyRea` demonstrates strong performance across mathematical reasoning tasks and generalizes well to other domains, several limitations remain. First, the effectiveness of latent reasoning depends on the quality of entropy-based step selection; inaccurate entropy estimates may lead to the compression of semantically important steps, resulting in performance degradation. Second, the current approach assumes access to well-structured Chain-of-Thought data, which may not be readily available in all domains. Third, although reinforcement learning improves the model's switching behavior, it introduces additional training complexity and computational cost. Lastly, while `HyReacv` reduces output length, it does not yet support interpretability of latent steps, which could hinder transparency in high-stakes applications. Future work may explore more adaptive selection mechanisms, improved interpretability of latent reasoning, and extensions to tasks with less structured reasoning formats.

