# OpenReview forum: "Learning to Reason over Continuous Tokens with Reinforcement Learning"
_ICLR.cc/2026/Conference — ICLR 2026 Poster_

### Official Review · Reviewer_DWUt · 2025-10-31

**Soundness:** 4
**Presentation:** 3
**Contribution:** 3
**Rating:** 8
**Confidence:** 3

**Summary:**

The authors propose aHybrid Reasoning framework that combines explicit (token-based) and latent (embedding-based) reasoning (i.e. interpretability vs efficiency) to tackle the inefficiency of standard CoT based reasoning. Training is a done as a two-step procedure.

1. Supervised Cold-Start phase :
the model starts to replace intermediate CoT steps (the ones with low entropy) with continuous embeddings. Specifically, for multiple rounds, such low entropy steps are identified, and replaced with ` [Latent] := <start-latent> c × [latent] <end-latent>`. The training loss is computed only on the explicit (non-latent) tokens.

2. RL phase:
The authors use GRPO to further train the model, with the standard accuracy and formatting reward. They additionally add a latent reward to encourage the model to think in latent space.

The training procedure is conducted on two mathematical reasoning datasets for the first phase, a third one added for the RL phase. The Qwen 2 family of models (7B-32B) is used. Evaluation is conducted on standard Math benchmarks.

Results show that the proposed approach strikes a better performance / efficiency tradeoff compared to the standard SFT and SFT+RL baselines. The authors moreover show that the entropy-based heuristic to identify which steps should be replaced in the first training phase performs well (better than random replacement), already striking a better tradeoff than standard SFT+RL. The authors additionally provide an analysis of different components of their method, as well as the transferability to other domains.

**Strengths:**

1. The authors are solving an important problem; the ability to retain the performance of CoT reasoning while making it more efficient has the potential for high impact in the field.
2. The ability for the model to learn what (and what not) to compress is interesting.

**Weaknesses:**

1. A discussion on the training cost (esp. the added number of forward passes and the additional training time required) is missing.

**Questions:**

1. How exactly is the latent reward computed ? the ratio of CoT steps that are latent ?
2. Can the model do a reasoning step partially in latent space ?
3. I am not sure I understood Figure 7. If $c$ is the number of latent tokens in each latent step, shouldn't having more latent tokens increase performance ? Or should we interpret this figure as, for a given CoT step that has e.g. 10 explicit tokens, we keep `10-c` as not latent and `c` as latent ?
4. In Figure 6, how are you measuring the entropy ? Are you only measuring it for the `<start_latent>` token ? or are you also measuring it for intermediate latent tokens ?

---

> ### Author Response · Authors · 2025-11-22
> **Response to Reviewer DWUt**
>
> Dear Reviewer DWUt,
>
> Thank you for your insightful reviews and comments. We appreciate the time and effort you have put into providing valuable feedback. We would like to address your concerns as follows:
>
> > Concern #1: Training cost
>
> The training cost for HyRea is relatively high, comparable to that of Coconut [1]. For instance, training the Qwen2.5-7B-Instruct model at the cold-start stage on 8 H200 GPUs takes approximately 40 hours. Furthermore, the RL fine-tuning phase in HyRea is on par with conventional RL methods, such as those employed in SimpleRL-Zoo [2].
>
> > Concern #2: Latent Rewards
>
> A reward of 1 is given if the generated answer includes the latent reasoning tags \<start_latent\>\<latent\>...\</end_latent\>; otherwise, the reward is 0.
>
> Additionally, we experimented with rewarding the presence of multiple \<latent\> segments (i.e., assigning higher rewards for more latent reasoning steps). However, this approach resulted in lower performance compared to our current setting, which only checks for the existence of at least one \<latent\> segment and leaves the extent of exploration to the model itself.
>
> > Concern #3: More \<latent\> worse performance
>
> While the result may not align with the most intuitive approach, increasing the number of <latent> actually makes training more difficult. The observed inverse relationship between **c** and performance indicates that the training stability of HyRea and hybrid reasoning degrades significantly as **c** increases.
>
> > Concern #4: Entropy
>
> Yes, you are right, we evaluate it on the \<start_latent\> token.
>
> [1] Training Large Language Models to Reason in a Continuous Latent Space. Arxiv 2024
>
> [2] SimpleRL-Zoo: Investigating and Taming Zero Reinforcement Learning for Open Base Models in the Wild. COLM 2025

---

### Official Review · Reviewer_CEcx · 2025-10-31

**Soundness:** 1
**Presentation:** 1
**Contribution:** 2
**Rating:** 2
**Confidence:** 4

**Summary:**

The paper presents HyRea, which is a hybrid reasoning framework for LLMs that learns to switch during inference between explicit, token‑based chain‑of‑thought and latent-space computation, marked by special start-end tokens. It is trained via an entropy‑guided cold‑start SFT that replaces low‑entropy CoT steps with latent spans while supervising only visible tokens, and GRPO‑based RL with rewards for accuracy, formatting, and latent use. They evaluate on math benchmarks MATH‑500, Minerva Math, AMC23, OlympiadBench with Qwen2.5‑7B/32B models, where HyRea achieves comparable accuracy with substantially fewer output tokens and exhibits a small number of learned mode switches per solution. It also yields shorter outputs with competitive performance on MMLU and GPQA, suggesting a practical path to more efficient multi‑step reasoning without sacrificing quality.

**Strengths:**

- The paper provides a hybrid policy through cold-start training with entropy‑based step replacement, and selectable number of latent tokens with switch tokens is novel on top of existing papers.

- Good ablation studies to show the importance of different components of the method, along with a  behavioral analysis of switching patterns.

- The results are competitive with fewer tokens on multiple math benchmarks.

**Weaknesses:**

- The paper has many under-specified details on the experiments, especially regarding the RL training and the fairness/properness of the baselines compared against, and these are not even specified in the appendix. This makes the work unconvincing in its current form. Also, there are some unclear or incorrect terminologies used in the paper. I'm willing to change my rating if I get satisfactory clarifications on experimental details.

- The paper states that latent reasoning operates directly on embeddings, but explicit decoding already operates on and outputs token embeddings. So, I believe this terminology through using the term "embedding" is not clear for describing HyRea, which uses the hidden state $h_t$ as input.

- > “reward consists of both a format reward and an accuracy reward, as well as a latent reward. Specifically, we use the latent reward to guide the model in generating the token `[Latent]`. Additionally, during loss computation, we exclude the `[Latent]` token and calculate the loss only over the remaining token steps.”

  The paper is vague in its description of the reward function and there are other details missing regarding the RL training.

- The token counts in 4.2 don’t match Table 1. Table 1 reports 387 tokens for HyRea‑7B on MATH‑500 (not 287), 425 on Minerva‑7B (not 372), and 369 for HyRea‑32B on MATH‑500 (not 269).

- The experimental results are only for Qwen2.5 7B and 32B models, and there are gaps in model sizes, especially smaller ones, and there is no discussion of how results would change with these sizes.

**Questions:**

- How is per‑step entropy measured—token‑level entropy aggregated per step? What thresholding is used during cold start? Also, how is the threshold on the number latent tokens $S$ is selected, does it depend on the dataset/task or length of the output? There's no discussion of this in the paper.

- It's not mentioned in the paper what explicit reward function, clipping, or group size ($G$) is used during RL training. More importantly, once a trajectory includes latent tokens through hidden states that are tied to the current model weights, doesn’t that violate the on‑policy constraint of GRPO?

- It's not stated in the paper what inference parameters are used across methods for the discrete tokens. Also, Section 2.1 is written based on argmax without sampling, which I believe is not the most general description of "explicit reasoning".

- The paper does not properly discuss the COCONUT paper's setting. How are you reproducing the COCONUT paper on math benchmarks? Because what COCONUT actually does is curriculum-based learning by replacing the discrete tokens one-by-one with latent tokens, and at the end, it does full latent inference. So, are you following this on some SFT dataset? If you're not doing this, then the comparison with COCONUT is not fair. Also, how are you extracting the answer from COCONUT/Soft-Thinking trajectories? Please clarify these.

- The paper also needs to discuss how they are reproducing Soft-Thinking paper's setting (as there are also some hyperparameters to tune, such as entropy in cold-stop during inference). Importantly, if COCONUT and Soft-Thinking are used without training, it's unfair to compare against them in Table 2, as they don't see any training data. This might be the reason why they are falling behind in the benchmarks.

- The paper is missing connection to existing works on reinforcement learning training using continuous tokens, some example of which are "Hybrid Latent Reasoning via Reinforcement Learning" (https://arxiv.org/abs/2505.18454) and "Continuous Chain of Thought Enables Parallel Exploration and Reasoning" (https://arxiv.org/abs/2505.23648). Please discuss how the proposed method compares to these previous works, and add them as baselines to the tables if necessary.

---

> ### Author Response · Authors · 2025-11-22
> **Response to Reviewer CEcx**
>
> Dear Reviewer CEcx,
>
> Thank you for your insightful reviews and comments. We appreciate the time and effort you have put into providing valuable feedback. We would like to address your concerns as follows:
>
> > Detail #1: Reward function
>
> We define our reward components as follows:
>
> - Format Reward: This follows the original Deepseek-R1 setting [1]. A reward is given if the answer adheres to the correct format. Specifically, if the reasoning is enclosed within \<think\>...\</think\> tags and the final answer is presented within \boxed{}.
>
>
>
> - Accuracy Reward: A reward of 1 is assigned if the answer is correct.
>
>
>
> - Latent Reward: A reward of 1 is given if the generated answer includes the latent reasoning tags \<start_latent\>\<latent\>...\</end_latent\>; otherwise, the reward is 0.
>
>   Additionally, we experimented with rewarding the presence of multiple <latent> segments (i.e., assigning higher rewards for more latent reasoning steps). However, this approach resulted in lower performance compared to our current setting, which only checks for the existence of at least one <latent> segment and leaves the extent of exploration to the model itself.
>
> > Detail #2: Training details
>
> - Loss and On-policy
>
> In our method, the latent span is an internal continuous computation rather than a sequence of labeled tokens. During SFT we therefore compute cross-entropy only on visible tokens, while supervising the special <start-latent> and <end-latent> markers (upweighted). In RL we use GRPO: trajectories are sampled on-policy from a frozen snapshot of the policy (the rollout policy). We then compute importance-weighted objectives on the discrete actions actually taken (including the start/end-latent markers), and we exclude the latent content from token-level likelihood terms. The latent computation still affects the reward and gradients indirectly through the decisions to invoke it and through the trajectory outcomes. We ensure the on-policy requirement by evaluating log-probabilities with the rollout policy and using clipped importance ratios to update the current policy.
>
> - RL training details
>
> For RL training, we primarily follow the default settings in the EasyR1 example configuration (examples/config.yaml), with the main change being that we disable the KL-divergence term.
>
> - Entropy replacement
>
> Given the cost of entropy computation, we precompute token-level entropies prior to training, aggregate them per step, and derive a fixed replacement order. We then train in stages, running 3 epochs per replacement level. Concretely: at epoch 0 no steps are replaced; at epochs 1–3, we replace the single lowest-entropy step; at epochs 4–6, we replace the two lowest-entropy steps; and so on, incrementally increasing the number of replaced steps according to the precomputed order.
>
> > Detail #3: Baselines
>
> For Coconut, we use the same training schedule as HyRea with a sequential replacement order. Specifically, at epoch 0 no steps are replaced; at epochs 1–3 we replace the first step; at epochs 4–6 we replace the first two steps; and so on, incrementally increasing the number of replaced steps. Because the replacement count matches HyRea’s settings, discrete tokens remain after training, resulting in a single switch between explicit and latent modes. For the softthinking, yes, we agree that they do not involve training so it is unfair. As recommended by reviewer wezr, we include CODI [2] as a baseline for comparison, as it also seeks to train latent CoT to reduce generation length.
>
> |         | Math-500 |         | Minerva Math |         | AMC23 |         | Olympiad Bench |         |
> | ------- | -------- | ------- | ------------ | ------- | ----- | ------- | -------------- | ------- |
> |         | Acc      | #Tokens | Acc          | #Tokens | Acc   | #Tokens | Acc            | #Tokens |
> | Coconut | 70.4     | 106     | 22.1         | 174     | 33.7  | 217     | 26.8           | 296     |
> | CODI    | 75.8     | 183     | 25.7         | 193     | 41.0  | 236     | 33.8           | 282     |
> | HyRea   | 83.6     | 387     | 27.2         | 425     | 48.2  | 526     | 39.6           | 583     |
>
> We observe that CODI outperforms Coconut in terms of performance while maintaining a comparable response length. Nonetheless, its performance remains inferior to that of HyRea.
>
> > Concern #1 Related work
>
> Thank you for suggesting these related works. We did not include them as baselines because they do not aim to compact answer length; instead, they leverage hidden states to enhance discrete reasoning. We appreciate the pointers and will incorporate these works into the related work and background sections.
>
> > Concern #2 Typo
>
> Thanks for pointing this out. We identified some statistical errors and updated the table, but forgot to revise the accompanying text. We have now corrected both accordingly.
>
> [1] DeepSeek-R1. Nature 2025
>
> [2] CODI: Compressing Chain-of-Thought into Continuous Space via Self-Distillation. EMNLP 2025

---

### Official Review · Reviewer_Nae9 · 2025-11-01

**Soundness:** 3
**Presentation:** 3
**Contribution:** 3
**Rating:** 6
**Confidence:** 4

**Summary:**

This paper focuses on making reasoning with LLMs efficient by utilizing both discrete and continuous chain-of-thought (CoT) tokens. Towards this, the paper proposes Hybrid Reasoning (HyRea) that enables the LLM to adaptively switch between discrete CoT and continuous CoT during generation. HyRea consists of two phase training involving 1) a supervised cold-start phase and 2) an RL phase. By evaluating HyRea on various standard benchmarks, the authors show that HyRea exhibits a favorable trade-off between the model quality/accuracy and efficiency in terms of thought token lengths. Furthermore, the authors also demonstrate the good out-of-domain generalization beyond mathematical tasks for HyRea.

**Strengths:**

- The paper studies a timely problem of making the LLM reasoning more efficient.
- The proposed HyRea method is simple and effective, showcasing a good trade-off between reasoning performance and thought token lengths across multiple benchmarks.
- The paper presents a detailed ablation study of various design choices.
- The paper explores the generalization ability of HyRea trained models beyond mathematical tasks.

**Weaknesses:**

- Given that HyRea is aimed at achieve a good performance vs. token length trade-off, which sometimes results into worse performance than SFT + RL with discrete CoT (see Table 1 & Table 3), it is important to consider agains baselines that focus on achieving similar trade-off with just discrete CoT. If one encourage shorter thought lengths in SFT + RL via some form of length regularization or training on compressed thought sequences, would the resulting performance vs. thought length trade-off be worse than HyRea?
- In Section 4.3, difference between **Random** and **Entropy** is not very significant. Authors should compare results from multiple runs to make a stronger case for **Entropy**.
- In L221-223, the authors don't take [Latent] tokens into account while defining loss. What is the justification for this? Did the authors perform ablation for this design choice?

**Questions:**

- Could the authors provide examples similar to Figure 4 but with longer CoTs? Do the HyRea produce CoTs remain interpretable?
- In Line 365, why are authors referring to CoT + RL as **upper bound** in accuracy? According to Table 2, this upper bounds is often exceeded by Random and/or Entropy.
- While Figure 5 shows superiority of Entropy for cold-start stage, looking at Table 2 and Figure 5, Random appears to be more amenable to RL as the gap between Random and Entropy after RL is quite small?
- In Line 409, the authors say "The inverse relationship highlights a trade-off...". According to Figure 7, smaller c is better for both accuracy and thought length. What trade-off are the authors referring to?
- In Section 3.1, the authors may want to highlight how their method differs from Coconut.
- In Line 40, the authors state ``However, reasoning over embeddings remains inherently
difficult and can lead to degraded model performance compared to traditional token-based inference,
as certain tokens encode complex, nuanced information that cannot be fully preserved in compressed
embeddings. `` Could the authors provide a citation or evidence for this?

---

> ### Author Response · Authors · 2025-11-22
> **Response to Reviewer Nae9**
>
> Dear Reviewer Nae9,
>
> Thank you for your insightful reviews and comments. We appreciate the time and effort you have put into providing valuable feedback. We would like to address your concerns as follows:
>
> > Concern #1: Compare with compacted discrete CoT
>
> Thank you for raising this important point. We are actively exploring this direction as well. Specifically, we have conducted experiments where the CoT is compacted during training by introducing a penalty reward on output length. However, we found that this approach underperforms compared to our proposed method. We compare our method with one of the most representative works in this area, L1 [1], and present the results below:
>
> |         | Math-500 |         | Minerva Math |         | AMC23 |         | Olympiad Bench |         |
> | ------- | -------- | ------- | ------------ | ------- | ----- | ------- | -------------- | ------- |
> |         | Acc      | #Tokens | Acc          | #Tokens | Acc   | #Tokens | Acc            | #Tokens |
> | SFT+RL  | 84.2     | 698     | 26.8         | 671     | 48.2  | 892     | 40.0           | 854     |
> | Coconut | 70.4     | 106     | 22.1         | 174     | 33.7  | 217     | 26.8           | 296     |
> | L1      | 84.4     | 556     | 26.1         | 464     | 47.0  | 602     | 40.2           | 646     |
> | HyRea   | 83.6     | 387     | 27.2         | 425     | 48.2  | 526     | 39.6           | 583     |
>
> From the results, we observe that HyRea consistently achieves competitive or superior accuracy while generating significantly more compact outputs, outperforming other methods that rely solely on discrete CoT for length reduction.
>
> > Concern #2: Random vs. Entropy
>
> Thank you for pointing this out. Indeed, the Random and Entropy-based settings achieve similar final accuracy after RL training, but differ in the number of tokens generated. Specifically, during the cold-start phase, as shown in Figure 5, Random converges significantly slower and performs worse than Entropy. However, after RL training, Random catches up in terms of accuracy but still produces longer answers compared to Entropy.
>
> > Concern #3: Loss
>
> We do not include \<latent\> in the loss calculation because it is not a real token, but rather a placeholder used to represent latent reasoning. It corresponds to the embedding rather than a discrete, labeled token. Since there is no ground-truth label for the content within \<latent\>, computing a loss over it is not applicable.
>
> > Concern #4: More examples
>
> Here is one concrete example of the longer CoT
>
> ```
>
> \<start_latent\>\<latent\>\<end_latent\>
>
> First, let me check if 196 is even. Since it ends with a 6, which is even, yes, it's divisible by 2. So dividing 196 by 2 gives 98. Now 98 is also even, so divide by 2 again: 98 ÷ 2 = 49. So far, the factors I have are 2, 2, and 49. So 196 = 2 × 2 × 49. But 49 is 7 squared, right? Because 7 × 7 = 49. So that means 196 = 2² × 7².
>
> \<start_latent\>\<latent\>\<end_latent\> Let me remember that formula. So if the prime factorization is p₁^a × p₂^b × ... × pₙ^k, then the number of divisors is (a+1)(b+1)...(k+1).
>
> Applying that here, the prime factorization is 2² × 7². So the exponents are 2 and 2. Adding 1 to each exponent gives 3 and 3. Multiplying those together: 3 × 3 = 9. So that means there should be 9 divisors in total. Let me verify this by listing them out.
>
> Starting from the prime factors: 2 and 7. The exponents are 2 each, so possible combinations:
>
> For 2: exponents 0, 1, 2
>
> For 7: exponents 0, 1, 2
>
> \<start_latent\>\<latent\>\<end_latent\>
>
> Let me count them: 1, 7, 49, 2, 14, 98, 4, 28, 196. That's 9 numbers. So yes, that matches the formula. Therefore, 196 has 9 positive whole-number divisors.
>
> Wait, just to be thorough, let me check if there are any other divisors. The prime factors are 2 and 7, so any divisor must be a product of these primes raised to some power. Since the exponents are 2 each, the possible combinations are exactly the ones listed above. So I think that's all. So 9 divisors.
>
> \<start_latent\>\<latent\>\<end_latent\>
>
> For example, 1 divides everything, so 1 is a divisor. 2 divides 196 because 196 is even. 4: 196 ÷ 4 is 49, which is an integer. 7: 196 ÷ 7 = 28. 14: 196 ÷ 14 = 14. 28: 196 ÷ 28 = 7. 49: 196 ÷ 49 = 4. 98: 196 ÷ 98 = 2. 196: 196 ÷ 196 = 1. So all these numbers divide 196. And there are no others between 1 and 196. So that's 9 divisors.
>
> \<start_latent\>\<latent\>\<end_latent\>\<start_latent\>\<latent\>\<end_latent\> \boxed{9}.
>
> ```
>
> > Concern #5: More rigorous statements
>
> Thanks so much for pointing out the question about several not quite rigorous statements. We have revised them in the paper.
>
> [1] L1: Controlling How Long A Reasoning Model Thinks With Reinforcement Learning. COLM 2025

---

### Official Review · Reviewer_wezr · 2025-11-01

**Soundness:** 3
**Presentation:** 3
**Contribution:** 2
**Rating:** 6
**Confidence:** 3

**Summary:**

The authors propose HyRea that lets LLMs adaptively alternate between explicit token-level reasoning and latent embedding-level reasoning, with special tokens marking the start and end of the latent span. The training has two stages:

1. Cold-Start Supervised Fine-Tuning (SFT):
Low-entropy CoT steps are replaced with latent spans to introduce embedding-based reasoning while preserving interpretability.

2. Reinforcement Learning (RL) with Group Relative Policy Optimization (GRPO):
Fine-tunes the model to learn optimal switching between reasoning modes, guided by task-specific rewards for accuracy, format, and efficiency.

Experiment results on math benchmarks, MMLU and GPQA show the performance of HyRea is in between vanilla SFT and SFT + RL, with significantly shorted generation.

**Strengths:**

1. The paper proposes a elegant solution of alternating latent and explicit reasoning with special tokens + cold start SFT + RL.

2. The performance of HyRea on math benchmark is close to vanilla SFT+RL, while the generation length is significantly shorter.

3. The paper is well-written and easy to follow.

**Weaknesses:**

1. There is only one latent CoT with short generation baseline (COCONUT), which was out last year, and is a weak baseline. The soft thinking baseline is training-free and does not shorten the generation, which does not align with the settings in the paper. It's a bit hard to calibrate the contribution of HyRea without suitable baselines. I'd suggest the authors to include more recent, stronger baselines that match their settings (training latent CoT to shorten generation). e.g. CODI (https://arxiv.org/abs/2502.21074). There are also quite a number of papers that produce mixed CoT chains. e.g. Token Assorted (https://arxiv.org/abs/2502.03275). RL with latent tokens: HRPO (https://arxiv.org/abs/2505.18454).
2. Please include a detailed description of the reward functions (accuracy, format, latent) in the main text.
3. If I understand correctly, the math-trained HyRea can perform reasonably well out-of-domain on MMLU and GPQA according to Table 3. Is SFT and SFT+RL also trained on math and tested OOD? It would be better to include other latent reasoning baselines to better calibrate the results.

**Questions:**

See weeknesses.

---

> ### Author Response · Authors · 2025-11-22
> **Response to Reviewer wezr**
>
> Dear Reviewer wezr,
>
> Thank you for your insightful reviews and comments. We appreciate the time and effort you have put into providing valuable feedback. We would like to address your concerns as follows:
>
> > Concern #1: More baseline
>
> Thank you for your recommendation. As you noted, we include CODI [1] as a baseline for comparison, as it also seeks to train latent CoT to reduce generation length.
>
> |         | Math-500 |         | Minerva Math |         | AMC23 |         | Olympiad Bench |         |
> | ------- | -------- | ------- | ------------ | ------- | ----- | ------- | -------------- | ------- |
> |         | Acc      | #Tokens | Acc          | #Tokens | Acc   | #Tokens | Acc            | #Tokens |
> | Coconut | 70.4     | 106     | 22.1         | 174     | 33.7  | 217     | 26.8           | 296     |
> | CODI    | 75.8     | 183     | 25.7         | 193     | 41.0  | 236     | 33.8           | 282     |
> | HyRea   | 83.6     | 387     | 27.2         | 425     | 48.2  | 526     | 39.6           | 583     |
>
> We observe that CODI outperforms Coconut in terms of performance while maintaining a comparable response length. Nonetheless, its performance remains inferior to that of HyRea. For the other works you recommended that generate mixed chains of thought, we will incorporate them into the discussion of related work and the background section of the main text.
>
> > Concern #2: Detailed description of reward function.
>
> We define our reward components as follows:
>
> - Format Reward: This follows the original Deepseek-R1 setting [2]. A reward is given if the answer adheres to the correct format. Specifically, if the reasoning is enclosed within \<think\>...\</think\> tags and the final answer is presented within \boxed{}.
>
>
>
> - Accuracy Reward: A reward of 1 is assigned if the answer is correct.
>
>
>
> - Latent Reward: A reward of 1 is given if the generated answer includes the latent reasoning tags \<start_latent\>\<latent\>...\</end_latent\>; otherwise, the reward is 0.
>
>   Additionally, we experimented with rewarding the presence of multiple \<latent\> segments (i.e., assigning higher rewards for more latent reasoning steps). However, this approach resulted in lower performance compared to our current setting, which only checks for the existence of at least one \<latent\> segment and leaves the extent of exploration to the model itself.
>
> > Concern #3: Out-of-domain tasks
>
> Yes, both the SFT and SFT+RL models are trained on mathematical tasks and evaluated on OOD settings. This evaluation protocol is consistent with the setup used by SimpleRL-Zoo [3].
>
> Additionally, we report the performance of various baselines under the same OOD setting:
>
> |         | MMLU |         | GPQA |         |
> | ------- | ---- | ------- | ---- | ------- |
> |         | Acc  | #Tokens | Acc  | #Tokens |
> | Coconut | 53.8 | 31      | 20.3 | 368     |
> | CODI    | 59.4 | 28      | 22.6 | 383     |
> | HyRea   | 68.6 | 53      | 27.4 | 685     |
>
> We observe that HyRea consistently achieves the best performance across both datasets, demonstrating superior generalizability to OOD tasks compared to other baselines.
>
> [1] CODI: Compressing Chain-of-Thought into Continuous Space via Self-Distillation. EMNLP 2025
>
> [2] DeepSeek-R1: Incentivizing Reasoning Capability in LLMs via Reinforcement Learning. Nature 2025
>
> [3] SimpleRL-Zoo: Investigating and Taming Zero Reinforcement Learning for Open Base Models in the Wild. COLM 2025

---

### Meta-Review · Area_Chair_fczj · 2026-01-07

**Summary:**

This paper introduces HyRea, a hybrid reasoning framework that enables LLMs to dynamically switch between explicit token-based and latent embedding-based reasoning during inference. The approach employs a two-stage training pipeline: an entropy-guided cold-start SFT phase that replaces low-entropy CoT steps with latent embeddings, followed by GRPO-based RL fine-tuning with task-specific rewards. The reviewers acknowledged the timeliness of efficient reasoning research, the elegance of combining special tokens with cold-start SFT and RL, and the favorable accuracy-efficiency trade-offs on mathematical benchmarks. The review scores are (6, 6, 2, 8), with the main disagreement centered on experimental rigor and baseline fairness. The authors provided substantive responses including additional baselines (CODI, L1), clarified reward functions and training procedures, and demonstrated competitive performance with reduced token counts. While not all concerns were fully resolved, the core contribution of learning to dynamically switch between reasoning modes with meaningful efficiency gains represents a useful advance, and the additional experiments strengthen the empirical support for the main claims.

**Reviewer Concerns:**

The authors addressed several key concerns during rebuttal. Reviewer wezr's request for stronger baselines was met with CODI comparisons, and the detailed description of reward functions was provided as requested. Reviewer Nae9's concern about comparison with compressed discrete CoT was addressed through L1 baseline experiments. The justification for excluding latent tokens from loss computation was clarified. For Reviewer CEcx, the authors provided specific details on training procedures, entropy measurement, and RL hyperparameters. However, some concerns remain partially outstanding. The fairness of COCONUT comparison was explained through matching training schedules, but the curriculum learning differences from the original paper were not fully reconciled. The authors acknowledged that Soft-Thinking comparison is unfair since it involves no training, yet this baseline remains in Table 2. Related works suggested by CEcx (HRPO, continuous chain-of-thought papers) were acknowledged for inclusion in discussion but not added as experimental baselines. The statistical significance of Random vs. Entropy-based replacement and experiments on smaller model sizes were not provided. Additionally, some typos in token counts (Table 1 vs. Section 4.2) and terminology issues should be corrected in the final version.

**Reviewer Scores:**

Reviewer wezr (Rating: 6) would likely maintain their score, as their concerns about baselines and reward descriptions were addressed, though contribution level remains a valid concern. Reviewer Nae9 (Rating: 6) would likely maintain their score, with responses to questions about discrete CoT compression and experimental details being satisfactory. Reviewer CEcx (Rating: 2) would likely increase their score moderately given the clarifications on experimental details and training procedures, though some concerns about baseline fairness and missing comparisons with recent related works remain unresolved; a revised score in the range of 4-6 seems plausible. Reviewer DWUt (Rating: 8) would likely maintain their positive assessment, as their questions about latent rewards, training costs, and methodology were adequately answered.

---

### Decision · Program_Chairs · 2026-01-26

Accept (Poster)